# PI3K/Akt Pathway: The Indestructible Role of a Vintage Target as a Support to the Most Recent Immunotherapeutic Approaches

**DOI:** 10.3390/cancers13164040

**Published:** 2021-08-11

**Authors:** Matteo Caforio, Emmanuel de Billy, Biagio De Angelis, Stefano Iacovelli, Concetta Quintarelli, Valeria Paganelli, Valentina Folgiero

**Affiliations:** 1Department of Pediatric Hematology and Oncology, Cell and Gene Therapy, IRCCS Bambino Gesù Children’s Hospital, 00146 Rome, Italy; matteo.caforio@opbg.net (M.C.); emmanuel.decrespin@opbg.net (E.d.B.); biagio.deangelis@opbg.net (B.D.A.); stefano.iacovelli@opbg.net (S.I.); concetta.quintarelli@opbg.net (C.Q.); valeria.paganelli@opbg.net (V.P.); 2Department of Clinical Medicine and Surgery, University of Naples Federico II, 80125 Naples, Italy

**Keywords:** PI3K, immunotherapy, PI3K/Akt inhibitors

## Abstract

**Simple Summary:**

PI3K/Akt pathway has an impressive story as tumor marker. PI3K-dependent solid tumors have been studied for several years in order to inhibit the pathway at different levels along the signaling. Despite the highly satisfactory results obtained in vitro and in xenograft mouse tumor models, the use of PI3K/Akt inhibitors in clinical trials resulted in being not as efficient as expected. With the emerging role of the tumor microenvironment in the response to therapy and the awareness, increasing in recent years, of the necessity to army the immune system against the tumor, new opportunities have emerged for PI3K/Akt inhibitors. Here, we show that PI3K/Akt, in addition to its function as tumor marker, exerts a pivotal role as an immunomodulator. Recent studies demonstrate that PI3K/Akt pathway is crucial for the regulation of the immune system and that its inhibition in combination with immunomodulatory agents may provide a new therapeutic approach for cancer.

**Abstract:**

Pathologic activation of PI3Ks and the subsequent deregulation of its downstream signaling pathway is among the most frequent events associated with cellular transformation, cancer, and metastasis. PI3Ks are also emerging as critical factors in regulating anti-tumor immunity by either promoting an immunosuppressive tumor microenvironment or by controlling the activity and the tumor infiltration of cells involved in the immune response. For these reasons, significant pharmaceutical efforts are dedicated to inhibiting the PI3K pathway, with the main goal to target the tumor and, at the same time, to enhance the anti-tumor immunity. Recent immunotherapeutic approaches involving the use of adoptive cell transfer of autologous genetically modified T cells or immune check-point inhibitors showed high efficacy. However, mechanisms of resistance to these kinds of therapy are emerging, due in part to the inhibition of effector T cell functions exerted by the immunosuppressive tumor microenvironment. Here, we first describe how inhibition of PI3K/Akt pathway contribute to enhance anti-tumor immunity and further discuss how inhibitors of the pathway are used in combination with different immunomodulatory and immunotherapeutic agents to improve anti-tumor efficacy.

## 1. Introduction

### 1.1. PI3K: Mechanisms of Action

Phosphatidylinositol 3-kinase/Akt-mammalian target of rapamycin (PI3K/Akt-mTOR) signaling pathways are active in several cellular compartments controlling multiple cellular processes such as cell proliferation, survival, differentiation, migration, and metabolism. Recent studies have shown that aberrant regulation of PI3K/Akt signaling is strongly involved in cancer and autoimmunity [1]. PI3Ks form an evolutionarily conserved family of lipid kinases whose members include Class IA subunits (p110α, β, δ), Class IB catalytic subunit (p110γ), Class I regulatory domains (p85α and β, p55α and γ, p50α, p101, p84, p87), Class II subunits (C2α, C2β, C2γ), Class III catalytic protein (Vps34), and regulatory protein (Vps15) [2]. In response to activating signals, PI3Ks are recruited to the plasma membrane where the catalytic subunit is dissociated from its regulatory subunit and becomes activated. Phosphorylation of Ptdlins (4,5) P2 by PI3Ks generates Ptdlins (3,4,5) P3, allowing for the recruitment and activation of the pleckstrin homology (PH) domain containing proteins such Akt that, in turn, phosphorylates and activates downstream signaling pathways [3]. Akt is a key component of this pathway and is directly implicated in the regulation of fundamental cellular processes, including survival, proliferation, and differentiation [4]. Dysregulation and activation of upstream effectors of the PI3K signaling pathway, such as RTKs and Ras mutations, or specific gene amplification and activating mutations in the PI3K alpha catalytic sub-unit (PIK3CA), as well as functional loss of PTEN, are all events frequently observed in many different cancer types [5]. Therefore, targeting PI3K with small molecules for PI3K-addicted tumor represents a valid therapeutic strategy. In addition, emerging translational work has identified hyperactivation of the PI3K pathway to be a significant mechanism of resistance to immunotherapy. Due to ubiquitous expression and involvement of PI3Ks in many biological processes, its role in shaping the tumor microenvironment and promoting a suppressive phenotype of immune cells has recently emerged [6]. In addition, specific PI3K isoforms are implicated in particular in regulating the fate and the activity of cells of the immune system. For instance, Class IA PI3Ks are involved in development and maturation of B cells and are an important mediators of antigen receptor signals [7]. In the T cell compartment, they play a major role in inducing and sustaining the expression of cytotoxic T lymphocyte effector molecules, such as perforin, IFNγ, and granzymes, as well as other biomarkers that distinguish memory and effector T cells [8]. In NK cells, PI3K sustains their degranulation activity, contributing to immune surveillance. Inhibition of PI3K in this cellular component of the innate immune response interferes with movement of perforin and granzyme B to target cells, indicating the crucial role of PI3Ks in the cytotoxicity of NK cells [9]. On the other hand, PI3K is critical for the function of Tregs, the T cell subset that contributes to impair immune response, secreting suppressive cytokines such as IL-10, TGF-β, and IL-35 that are able to inhibit the function of effector T cells. PI3K is involved in the immune suppressive activity of Treg cells, but its role appears contradictory. Indeed, PI3K inhibitors could be used to induce Treg differentiation; however, at the same time, inhibition of PI3K signaling in mice does not lead to increase the numbers of Treg. Therefore, inhibiting PI3K can facilitate the differentiation of Treg in vitro, but in vivo it reduces the number of Tregs and diminishes, without abolishing, their suppressive capacity [10] (Figure 1).

Because of the involvement of mainly the Class I PI3Ks in both tumorigenesis and immunosuppression, intense efforts have been dedicated to the development of selective isoform-specific PI3K inhibitors to be used in clinical settings, either as single drug or in combination with other immunomodulatory or immunotherapeutic agents.

In this review, we summarize the latest advances inherent to how PI3K inhibitors can contribute to enhance antitumor immunity and further discuss how they are used in combination with different immunomodulatory and immunotherapeutic agents to improve efficacy.

### 1.2. Recent Approaches of Anti-Tumor Immunotherapy

Due to the dual role of PI3Ks in tumor and immune compartments, new therapeutic approaches combining PI3K inactivation with most recent immune check-point inhibitors, adoptive cell transfer therapy, and immunomodulatory agents have been developed. The blockade of PD1/PD-L1 and CTLA4/CD80 or CD86 is the most common strategy to exert check-point inhibition [11]. In response to inflammation, PD1 expression is induced on the surface of the T cells and binds PD-L1 present on the tumor cells, enabling T cell-mediated tumor cell death. The same mechanism of action is described for CTLA4 and its interaction with the ligands CD80 and CD86. mAbs able to interfere with PD1 or CTLA4 and their respective ligands have been developed with the goal to restore T cell cytotoxic activity [12]. The adoptive cell transfer therapy is an innovative immunotherapeutic approach that implies the genetic engineering of T cells collected from the blood of a patient with a CAR construct targeting a specific antigen expressed at the surface of the tumor cells. Once engineered, CAR T cells are then re-infused to the same patient. On the basis of this approach, CAR T cells targeting the antigen CD19 have produced unprecedented clinical results for B cell malignancies, leading to its expedited FDA approval [13]. On the basis of these clinical successes, a new generation of CAR T cells directed against different types of tumor antigens are under development [14]. In addition, several small chemicals demonstrating anti-tumor activity have recently also been shown to have immunoregulatory properties.

The number of tumor-infiltrating T cells, deregulation of check-point molecule expression in tumor and T cells, mechanism of resistance exerted by factors of the tumor microenvironment, and mechanisms of relapse upon CAR T infusions are emerging as important limiting factors for immunotherapy. These have led to the development of several combinatorial therapeutic approaches. In light of these considerations, this review focuses on new insights into how PI3K inhibitors can contribute to enhance antitumor immunity and their application in combination therapy can improve the outcome of immunotherapy.

## 2. PI3K Inhibition in Combination with Immunomodulatory Agents

As single agent, the use of PI3K inhibitors for the treatment of a variety of cancers have thus far demonstrated limited efficacy, probably because there is not enough drug to have a significant biological activity and to observe a significant therapeutic response at the doses currently employed. Increasing the dose results into severe toxicities as PI3K signaling pathway is required for the homeostasis of normal tissues [15]. Decreasing the dose of PI3K inhibitors and combining them with other targeted therapies or immunotherapeutic agents could be beneficial by both limiting the toxicity and increasing efficacy [16]. Such combinatorial strategies may involve the use of a specific PI3K inhibitor, together with immune checkpoint blockade or small molecules targeting pathways involved in immunomodulation.

### 2.1. Immune Check-Point Blockade in Combination with PI3K Inhibitor

One of the best examples for the use of a PI3K pathway inhibitor in combination with immune checkpoints blockade involves PD1. Indeed, available data demonstrate that PD-L1 expression, in many different tumor cells, is related to the PI3K pathway. For example, both PTEN-mutant triple negative breast cancer (TNBC) and colorectal cancer (CRC) have higher expression of PD-L1, and PI3K inhibition induces a reduction in PD-L1 expression in tumor cells [16,17]. In particular, in TNBC, it was demonstrated that there is a correlation between PD-L1 degradation and Akt-mediated GSKβ inhibition. In fact, it was found that a binding domain on PD-L1 is recognized by GSKβ, which can induce degradation of PD-L1. Since GSK is negatively regulated by Akt, the inhibition of PI3K/Akt pathway and, therefore, the activation of GSKβ induces a reduction in PD-L1 protein level, providing important perspective for combinatory therapy [18]. Other examples show a correlation between the overexpression of PI3K pathway and PD-L1 expression. It is likely that the PI3K/Akt pathway regulates PD-L1 expression by either transcriptional or post-transcriptional mechanisms in a cell and tissue-type-dependent manner [19]. PTEN, a well-known biological inhibitor of PI3K pathway, results in mutation in some kinds of cancer and its function results abrogated. Therefore, in this condition, PI3K pathway is hyper-activated and, consequently, tumor cells exhibit high expression of PD-L1. A good example of PTEN-mutated tumor is melanoma. The anti-tumor activity of check-point blockade combined with the PI3Kβ inhibitor in xenograft mouse model of melanoma was tested. Treatment with each single agent had minimal effect, but PI3Kβ inhibitor in combination with an anti-PD1 antibody significantly improved tumor growth inhibition and survival of the mice [20]. Another example shows the blockade of an upstream regulator of the PI3K pathway, namely, HER3. HER3 blocking antibody treatments exert a potent antitumor effect by suppressing HER3–PI3K–Akt–mTOR oncogenic signaling, reducing the immunosuppressive microenvironment and preserving T cell activity [21]. Therefore, HER3 inhibition and PD1 blockade may provide a multimodal precision immunotherapeutic approach [21].

In melanoma xenograft in vivo model, PI3Kβ inhibition showed synergistic effects in combination with CTLA4 inhibition. Of note, the simultaneous inhibition of PI3Kβ and check-point signal augmented the number of the infiltrating CD4+ and CD8+ T cells in all the models analyzed. Collectively, these data suggest that PI3Kβ inhibition can be exploited to improve the efficacy of immunotherapy in melanomas characterized of PTEN loss [20]. Several trials are now using a drug combination to target the PI3Kβ pathway and immune check-point inhibitors alone (NCT03131908) or in combination with other inhibitors (NCT03772561). In particular, a phase I/II trial is testing a PI3K inhibitor with the PD1 and CTLA4 inhibitors nivolumab and ipilimumab (NCT04317105) in PI3K/Akt-mutated solid tumors. Other trials are exploring the combination of Nab-rapamycin and nivolumab in advanced sarcoma (NCT03190174) [22]. Encouraging data are provided by studies regarding non-small cell lung cancer. In this context, activation of Akt and mTOR is associated with PD-L1 expression, and inhibition of PI3K, Akt, and mTOR activity induces the decrease of PD-L1 expression. The combination of rapamycin and anti-PD1 significantly reduces lung tumor burden in comparison with any other treatment group [23] (Figure 2).

### 2.2. Immunomodulatory Drugs in Combination with PI3K Inhibitors

PI3K inhibitors are also used in combination with other small molecule inhibitors, which possess immunomodulatory activities and target key tumorigenic drivers. These strategies may present the double advantages to increase both tumor cell death and anti-tumor immunity.

The Bruton tyrosine kinase (BTK) inhibitor ibrutinib was originally developed for treatment of chronic lymphocytic lymphoma (CLL). Both BTK and PI3K pathways are activated by the B cell receptor in CLL, and co-inhibition of BTK and PI3K delta (idelalisib) in preclinical model of aggressive lymphomas is reported to significantly improve the efficacy [24]. Ibrutinib has recently been shown to improve T cell function in CLL, resulting in the expansion of memory T cells, TH1 polarization reducing expression of inhibitory receptors, and improved immune synapse formation between T cells and CLL cells [25,26]. While ibrutinib may act through BTK-dependent and -independent mechanisms [25], it is likely that the anti-CLL effect observed with the co-administration of ibrutinib and idelalisib is mediated via the direct inhibition of both pathways in tumor cells and potentially through the modulation of the immune response.

Other drugs, such as the E3 ligase cereblon inhibitor thalidomide, or lenalidomide, as well as the proteasome inhibitor (Proteasome inhi) carfizomib, have demonstrated immunomodulatory functions. Genetic ablation of cereblon in mice or its inhibition with thalidomide increases T cell anti-tumor immunity by restoring the metabolic activity of the effector T cells in an immunosuppressive and hostile tumor microenvironment [27], and carfizomib enhances NK cell lytic activity [28]. Their use in combination with specific selective PI3K inhibitors have demonstrated synergistic cytotoxicity against diffuse large B cell lymphoma and multiple myeloma, respectively [29,30].

BRAF and MAPK are others important signaling pathways particularly implicated in the progression of melanoma and thyroid cancer. Frequent alteration of the *BRAF* gene is observed in 80% of melanoma and about 45% of sporadic papillary thyroid cancers [31]. While BRAF is a key driver in these tumor types, it is now well established that BRAF signaling plays an important role in immunosuppression [32,33,34]. For example, oncogenic BRAF suppresses T cell activation in part through the induction of immunosuppressive cytokines, and increase of PD-L1 expression [35] and BRAF inhibition enhances T cell infiltration. The dual inhibition of the BRAF/MEK and PI3K signaling pathways has been shown to substantially improve anti-tumor activity in different tumor types, essentially by inhibiting cross talks between the two pathways, which are responsible for resistance mechanisms [36]. Moreover, the demonstration that co-targeting BRAF/MEK and PI3K enhances the anti-tumor efficacy of PD1 blockade in melanoma indicates that BRAF and PI3K co-inhibition acts not only through direct induction of tumor cell death, but also indirectly by enhancing anti-tumor immunity (Figure 2) [37].

## 3. Modulation of PI3K to Improve CAR T Therapy

In recent years, T cells genetically modified with a CAR have achieved unprecedented and impressive clinical results in patients with large B cell lymphoma or B cell acute lymphoblastic leukemia (ALL) [14,38]; however, available data indicate only limited efficacy of the CAR T cell approach in patients with solid tumors [39,40,41,42,43,44]. Structurally, traditional CARs are defined by four elements: a single-chain variable fragment (scFv), designed to recognize the tumor antigens; a hinge and transmembrane domain, to anchor the chimeric receptor on the cell surface; and intracellular signaling domains that activate T cells following antigen engagement. The intracellular signaling domain is classically constituted by the T cell activation domain CD3-ζchain (first generation). The additional introduction of either one or two costimulatory molecules (i.e., CD28, 4-1BB, OX40, ICOS, CD27, or other) in the construct is typical of second- and third-generation CARs, respectively [45,46].

Several investigators have worked on the structure of the CARs to improve their tumor infiltration, reduce the tumor escape, and to render them resistant to the immunosuppressive tumor microenvironment (TME). Control of T cell activation and differentiation by PI3K is particularly relevant to CAR T cell immunotherapy [47]. Therefore, some authors investigated the possibility to develop a combinatorial therapy using PI3K inhibitor and CAR T cells in order to ameliorate their performance and to prolong their persistence. Multiple studies now indicate that modulation of PI3K and its downstream targets is a promising approach to improve CAR T cell efficacy by limiting CAR self-signaling effects and improving T cell memory formation, survival, and function. Optimizing the use of inhibitors of these pathways for clinical application is the next challenge. Recent experiments performed in acute myeloid leukemia (AML) model showed that inhibition of p110δ PI3K has also been found to enhance efficacy and memory in tumor-specific therapeutic CD8 T cells, while inhibition of p110α PI3K increased cytokine production and antitumor response [48]. Then, sustained Akt activation leads to T cell terminal differentiation but its inhibition in CAR-modified T cells results in an early memory phenotype and improved antitumor efficacy [49].

Generally, for the treatment of AML, CAR T cells targeting CD33 antigen, a transmembrane receptor highly expressed on leukemia cells, have been developed. The efficacy of this cell-based therapy is sustained by in vivo experiments involving a mice model. In this murine model, the efficacy of PI3K inhibition was tested in combination with the infusion of CD33.CAR T cells. At day 14 after transfer, it was observed that the persistence of CD33.CAR T cells in liver and spleen was improved by the administration of PI3K inhibitors LY294002 (LY), resulting in reduced tumor burden. The addition of LY is also able to maintain the less differentiated phenotype of CAR T cells and to enhance the total cell number. The anti-tumor efficacy is needed to be further improved to find better experimental conditions for PI3K inhibition [50]. Other authors have focused their attention on the downstream PI3K effector Akt in order to improve CAR T cell efficacy. In particular, interesting data have been provided by experiments involving CAR T cells directed to epithelial cell adhesion molecule (EpCAM-CAR T). In this context, Akt inhibition does not suppress proliferation or effector function but increases the number of CAR-positive cells and inhibits the terminal differentiation of the EpCAM-CAR T cells. Furthermore, the EpCAM-CAR T cells, expanded in the presence of Akt inhibitor (Akti) MK2206, appeared to have enhanced antitumor activity in vivo. Taken together, these findings suggest that Akt inhibition during the initial stage of CAR T cell preparation could improve the performance of CAR T cells. These studies demonstrate that the Akti does not suppress EpCAM-CAR T cell expansion, viability, or effector functions but promotes the generation of memory EpCAM-CAR T cells with enhanced CAR-positive expression rates and greater antitumor activity in an in vivo model [47]. Moreover, it has been suggested that the sustained activity of Akt induces terminal differentiation in T cells and leads to decreased antitumor activity. Therefore, it was hypothesized that the inhibition of Akt may prevent terminal differentiation of CAR T cells as they are produced to a clinical dose. To evaluate this hypothesis, they performed an experiment of transduction and expansion of CD19 CAR T cells in the presence of an Akti. Akt inhibition during ex vivo expansion resulted in a generation of less differentiated CAR T cells, characterized by high expression of CD62 and CD28, with enhanced antitumor activity in vivo. Finally, it has been suggested that the modulation of Akt could be a good strategy to increase the anti-tumor immunity for adoptive CAR T cell therapy, which could be translatable into the clinical activity with the availability of a pharmaceutically approved Akti [51] (Figure 3).

### Inhibition of PI3K/AKT Pathway in CAR T Cell Manufacturing

CAR T cells are ‘living drugs’ with the capacity to migrate, proliferate, and kill cognate antigen-expressing target cells in vivo. Despite variation in practices and technology between the different production centers, all CAR T cells for clinical use are manufactured following the GMP rules and share the following production steps: activation, gene delivery (for an efficient receptor engineering of human T cells), expansion, and final product formulation. An average of 12 days is required for GMP CAR T cell manufacture [52,53]; however, in order to prevent exhaustion phenotype, studies are now testing the possibility of reducing the ex vivo processing time [54]. Generation of T central memory (T_CM_) phenotype cells remains one of the principal goals to improve T cell therapies [55]. Of note, many of the current clinical manufacturing methods for CAR T cells are focused mostly on producing as many cells as possible [56]. In particular, this is the principal aim when apheresis from heavily pretreated patients that contain a low level of naive/early memory T cells is used as starting material [57,58,59].

In their study, Klebanoff et al. suggest a new strategy for the manufacturing of CAR T cells using Akt inhibition, with the final goal of producing a large number of cells with an early memory phenotype [60]. To perform this study, they started from an unfractionated population of peripheral blood mononuclear cells (PBMC) and performed all the manufacturing steps (T cell activation and retroviral transduction of a second-generation anti-CD19 CAR) in the uninterrupted presence of AKTi (AKT inhibitor VIII) or, as control, the vehicle (Veh). Their results showed no difference in the cells grown in the presence of AKTi compared with Veh control, in terms of growth rate kinetics and overall expansion of anti-CD19 CAR-modified cells. Furthermore, they showed that the levels of transduction, either in CD4+ or CD8+ T cells, remained substantially unchanged when used the Akti during the manufacturing. Interestingly, they found that presence of the Akti significantly preserved CD62L expression on the CD19 transduced T cells compared with the control (Veh). Of note, CD62L plays a pivotal role as a key marker of highly potent T cell subsets for adoptive immunotherapy [61,62,63], and this role is Akt 1/2-dependent [64]. They finally showed that a manufacturing process in the presence of Akti leads to an efficient activation, expansion, and retroviral transduction of human T cells with a CAR, promoting the simultaneous expression of CD62L. Moreover, they showed that the generation of a population of human CD62L+ T_CM_-like cells, in the presence of Akti, rely on a FOXO1-dependent mechanism. Summarizing, they concluded that the capacity of Akti to allow the CAR T cell production, with the preserving of a low differentiated CD62L-expressing cells, was associated with conserved MAPK signaling pathway and the intranuclear accumulation of FOXO1. Interestingly, on the basis of these results, a human gene therapy clinical trial, which involves the use of the Akti in the T cell manufacturing process, was opened to enrollment (NCT0313937) [60].

It has been demonstrated that the terminal differentiation of CD8+T cells is promoted by the constitutive activation of PI3K/Akt-mTOR, while the inhibition of this pathway promotes T cell memory development [50,65,66]. In their work, Zheng et al. have shown that tonic CAR signaling promotes the constitutive activation of the PI3K/Akt pathway. This occurs in concert with the persistent ligand-independent activation of TCR CD3ζ signaling through the CAR [50]. They found that tonic CAR signaling during the manufacturing, in particular the CAR T cell expansion step, reduced T(naïve)_N_/T(stem central memory)_SCM_ and increased T(effector)_EFF_ subsets that correlate with poor persistence of the CAR T cells in vivo. The data obtained in this study show that the PI3K inhibition during CAR T cell manufacturing, particularly in the ex vivo expansion, increases T_N_/T_SCM_ and T_CM_ populations, and decreases the T_EFF_ population, resulting in improved in vivo persistence and cytokine production of the CAR T cells [60]. They finally suggest incorporating the PI3K inhibitor treatment into production regimens, in particular during the ex vivo CAR T cell expansion, where it is associated with unperturbed cell growth and without impacting on finally yield. In a recent publication, Dwyer et al. report that PI3K inhibition, during the manufacturing, may improve the efficacy of CAR T cells in vivo [67]. Using T cells isolated from peripheral blood as starting material, they showed how the ex vivo inhibition of either PI3Kγ or PI3Kδ, but not simultaneous inhibition of both subunits, generates T cells with enhanced therapeutic capabilities in vivo. Conversely, they observed that T cells treated with dual inhibitor IPI-145 were less effective at regressing tumors than T cells treated with more selective drugs targeting only PI3Kγ or PI3Kδ. They further showed that PI3Kδ blockade was more effective than PI3Kγ inhibition for the generation of human CAR T cells with enhanced lytic capacity in vitro, suggesting that PI3Kδ is the best therapeutic target, for the inhibition treatment, in the PI3K/Akt pathway. Moreover, performing an in vivo study, they observed that the majority of PI3K-inhibited T cells in the blood of mice showed a central memory phenotype (CD44^+^CD62L^+^). Several studies have shown that CAR T cell persistence is associated with in vivo efficacy and sustained disease remission [68,69,70,71,72]. All the studies here reported show that CAR T cell differentiation, exhaustion, and metabolic status affect the survival and efficacy of adoptively administered cells. Albeit more detailed studies are needed, but the inhibition of PI3K/Akt pathway during ex vivo expansion seems to be a promising and encouraging strategy for producing T cells with a less differentiated state (T_N_/T_CM_ phenotype) and improving their in vivo persistence and anti-tumor efficacy (Figure 4).

## 4. Classification of PI3K Inhibitors Used in Combined Treatments

Different isoform selective PI3K inhibitors have been developed to treat diseases. To date, five of them are approved by the FDA and/or EMA agencies either as a single agent or in combination for the treatment of different malignancies. With the exception of the selective PI3K alpha inhibitor alpelisib, approved in combination with fulvestran for HR-positive and HER2/neu-negative breast cancer therapy [73], the other four remaining drugs are indicated for various hematological malignancies. Idelalisib, a specific PI3Kδ inhibitor, in combination with an anti-CD20 antibody, rituximab [74], as well as duvelisib, a dual PI3Kδ/γ selective inhibitor [75], are both approved for the treatment of relapsed or refractory CLL and small lymphocytic and follicular lymphomas. Copanlisib, which targets the PI3Kα/δ isoforms, is indicated for patients affected by follicular lymphoma [76], and umbralisib, a dual PI3Kδ and casein kinase 1 inhibitor, was recently approved for patients affected by relapsed or refractory marginal zone lymphoma who previously received and anti-CD20 therapy and relapsed and refractory follicular lymphoma [77].

Other compounds have either entered early clinical stages or received a fast-track designation with the FDA. For instance, the Pan-PI3K inhibitor buparlisib entered into a phase 3 clinical trial for head and neck cancer patients who received a prior treatment with anti-PD1 therapy (NCT04338399; see Table 1). Zandelisib, a selective PI3Kδ inhibitor, also entered recently into phase 3 in combination with rituximab for patients affected by non-Hodgkin, follicular, and marginal zone lymphomas (NCT04745832; Table 1).

While these PI3K inhibitors have been shown to be effective by demonstrating promising results in the clinic, their uses are associated with several adverse effects, and a plethora of clinical trials for testing these agents in combination are ongoing to improve efficacy and safety. Moreover, the immunomodulatory function of PI3Kδ and γ isoforms in particular has opened new opportunities for their use in combination with other immunotherapeutic or immunomodulatory agents (Table 1).

Several early and late-phase clinical trials involving antibodies against CD20 (rituximab) for B cell malignancies, or against ErbB2 for HER2-positive breast cancers in combination with PI3K inhibitors are demonstrating promising therapeutic improvement. For examples, copanlisib, zandilisib and umbralisib in combination with rituximab, or alpelisib in combination with trastuzumab or pertuzumab are now active and/or recruiting for phase 3 clinical trials (Table 1).

Another strategy in clinical testing implies the use of a PI3K antagonist in combination with immune checkpoint inhibitors. The rationale for such a combination is that PI3K inhibitors will inhibit tumor cell proliferation, as well as, through their immunomodulatory functions, act in concert with an immune checkpoint inhibitor to help the body’s immune system attack cancer cells. While several clinical trials are ongoing using inhibitors with different selectivity toward the PI3K isoforms and antibodies targeting PD1 (Table 1), it is, however, important to note that two different trials using pembrolizumab in combination with selective dual PI3Kδ/γ inhibitor duvelisib for the treatment head and neck cancer and tenalisib for relapsed or refractory cHL have been suspended or terminated in phase 1/2 clinical trials (NCT04193293, NCT03471351; Table 1).

Several early stages clinical trials using PI3K inhibitors in combination with other small chemical drugs with immunomodulatory functions are ongoing. For example, lenalidomide, an inhibitor of cerebelon E3 ligase, or the selective proteasome inhibitor carfilzomib, both known for their immunomodulatory function [78,79,80], are in early stages clinical testing in combination with umbralisib for the treatment of patients affected by Hodgkin’s and mantle cell lymphomas. Vemurafenib, a Braf inhibitor described to increase the production of immune stimulatory cytokines and decrease immunosuppressive cytokines levels [32], is in early phase clinical testing in combination with the selective PI3Kδ/γ antagonist copanlisib for patients affected by thyroid tumors (NCT0446271, Table 1). However, a similar study using vemurafenib in combination with the pan-PI3K inhibitor buparlisib for the treatment of advanced melanomas has been stopped as has not been tolerated in patients [81].

## 5. Conclusions

The effect of tumor immunotherapy is influenced by many factors including the immune suppressive status of the patient and the intratumor heterogeneity. Cell immunotherapy is now effective for some hematological malignancies, but the efficacy is still limited within solid tumors, mainly due to the tumor heterogeneity and microenvironment. The more specific and personalized approach with adoptive cell transfer of CAR T cells showed strong targeting, small side effects, and the ability for it to be used in some advanced patients or patients with no response to other curative effects. Although anti-tumor immunotherapy is playing an increasingly important role and encouraging results have been obtained in clinical trials for various malignant tumors, it is necessary to reduce adverse reactions and to improve the efficacy by finding new targets and new strategies such as combination therapy. The PI3Ks, regulating a broad range of downstream molecular effectors, when mutated or overexpressed, are able to control the oncogenic transformation of tumor cells and the progression of a variety of tumors in vivo. Furthermore, it is required in multiple processes, including not only cancer progression but also escape of cancer cells from immunological surveillance and immune suppression. Since tumor growth may be the result of tumor proliferation and tumor-induced failure of immunity in killing cancer cells, the pharmacological inhibition of PI3Ks is being revealed as beneficial because of both blockage of tumor growth and immunosuppression of the tumor microenvironment.

The future of cancer therapy should include different approaches: the recovery of specific immune immunosuppressive pathways in the anti-tumor process in order to understand the mechanisms of action active in the tumor microenvironment and the cooperation with target therapy with a consolidated efficacy as anti-cancer drugs, such as PI3K inhibitors, exceeding the limits of the treatment as single agent.

## Figures and Tables

**Figure 1 cancers-13-04040-f001:**
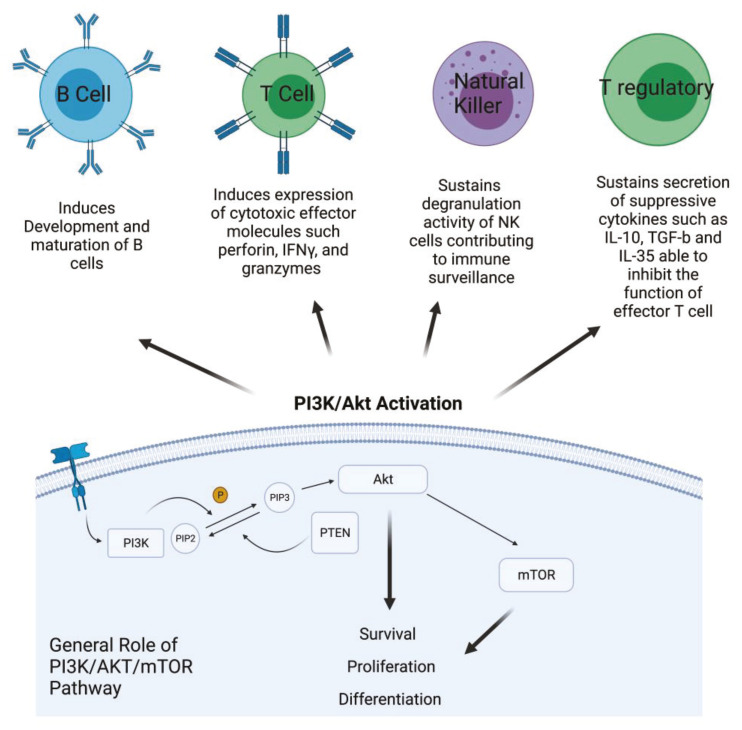
PI3K/Akt activation in cancer cells and its role in different kind of immune cells. PI3K is activated by growth factor receptors and, through the phosphorylation of PIP2 in PIP3, activates AKT that, in turn, activates mTOR. The effects of this activation involve survival, proliferation, and cell differentiation. Description of the effect of PI3K/Akt pathway activation in the main compartment of the immune system (Created with BioRender.com: agreement number RH22NUG37V).

**Figure 2 cancers-13-04040-f002:**
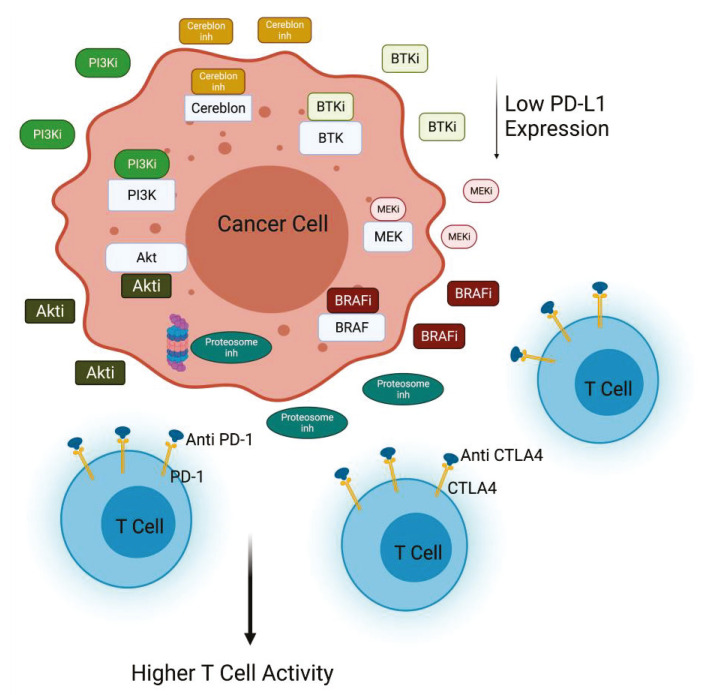
PI3K/Akt inactivation combined with immunomodulatory agents. Inhibition of PI3K or Akt decreases PD-L1 expression in cancer cells, contributing to the efficacy of anti-PD1 treatment. PI3K inhibitors in combination with different inhibitors (anti-CTLA4, BRAFi, BTKi, MEKi, Proteasome inhi) improves T cell killing activity against cancer cells (Created with BioRender.com: agreement number ZU22NUG38X).

**Figure 3 cancers-13-04040-f003:**
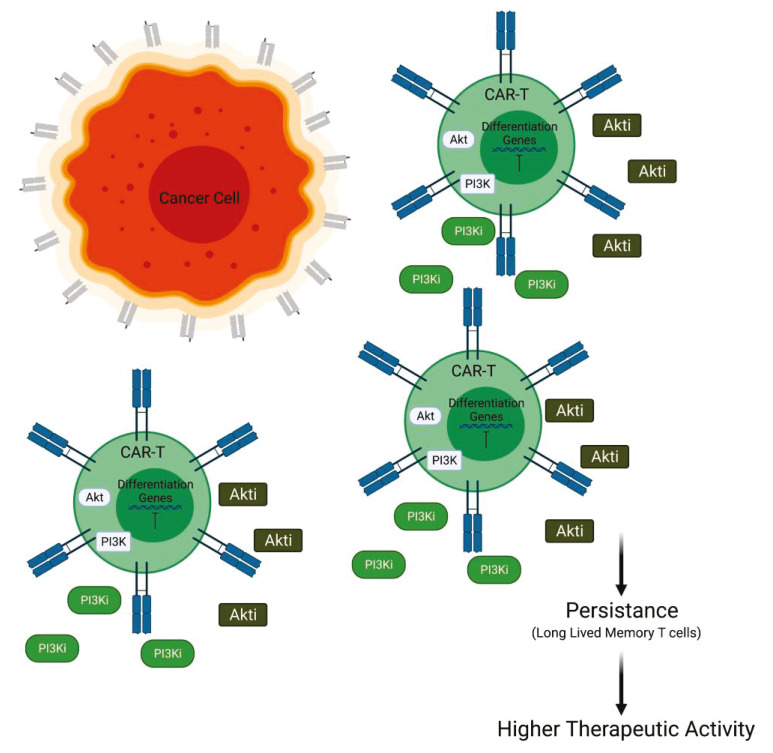
PI3K/Akt inactivation combined with CAR T cell therapy. Inhibition of PI3K or Akt is able to prolong the persistence of CAR T cell blocking the differentiation versus short memory T cells and promoting long-lived memory T cells (Created with BioRender.com: agreement number RXM22NUG34R).

**Figure 4 cancers-13-04040-f004:**
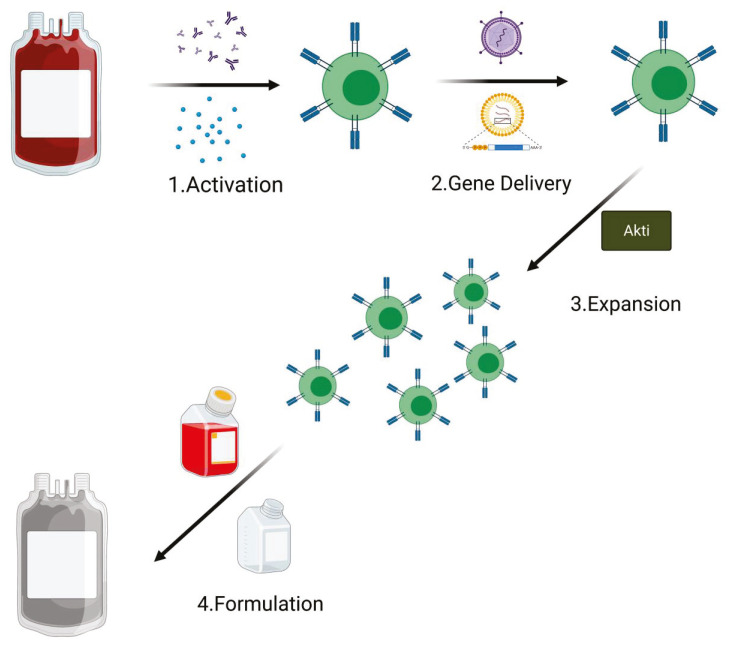
PI3K/Akt inactivation during the CAR T cell manufacturing process. The addition of Akti in the expansion phase sustains amplification of transduced T cells. The addition of Akti during CAR T cell expansion in the manufacturing process improves its expansion and efficacy (Created with BioRender.com: agreement number PT22NUG36Q).

**Table 1 cancers-13-04040-t001:** Classification of immunomodulatory inhibitors used in clinical trials.

Drugs Name	Clinical Trial Number	Combination Agents	Targets	Phase	Histopathology	Status
*PI3K delta*/*CK1* inhibitor						
** Umbralisib (TGR-1202)	NCT03269669	Obituzumab	Anti-CD20	2	Follicular lymphoma (grade (I-IIIa)	Recruiting
** Umbralisib (TGR-1202)	NCT03919175	Rituximab	Anti-CD20	2	Follicular lymphoma and marginal zone lymphoma	Recruiting
** Umbralisib (TGR-1202)	NCT02793583	Ublituximab	Anti-CD20	2b	Non-Hodgkin’s lymphoma	Recruiting
** Umbralisib (TGR-1202)	NCT02793583	Ublituximab + bendamustine	Anti-CD20/metabolism	2b	Non-Hodgkin’s lymphoma	Recruiting
** Umbralisib (TGR-1202)	NCT03801525	Ublituximab	Anti-CD20	2/3	Chronic lymphocytic leukemia	Recruiting
** Umbralisib (TGR-1202)	NCT03801525	Ublituximab + venetoclax	Anti-CD20/BLC2	2/3	Chronic lymphocytic leukemia	Recruiting
** Umbralisib (TGR-1202)	NCT04016805	Ublituximab + ibrutinib	Anti-CD20/BTK	2	Chronic lymphocytic leukemia	Recruiting
** Umbralisib (TGR-1202)	NCT04016805	Ublituximab + acalabrutinib	Anti-CD20/BTK	2	Chronic lymphocytic leukemia	Recruiting
**Umbralisib (TGR-1202)	NCT04016805	Ublituximab + venetoclax	Anti-cd20/BCL2	2	Chronic lymphocytic leukemia	Recruiting
** Umbralisib (TGR-1202)	NCT03283137	Pembrolizumab	Anti-PD1	1	Relapsed/refractory CLL and B cell non-Hodgkin’s lymphoma	Recruiting
** Umbralisib (TGR-1202)	NCT03379051	Ublituximab + lenalidomide	Anti-CD20/cereblon	1/2	Relapsed/refractory CLL/SLL and B cell non-Hodgkin’s lymphoma	Recruiting
** Umbralisib (TGR-1202)	NCT03379051	Ublituximab + vetetoclax	Anti-CD20/BCL2	1/2	Relapsed/refractory CLL/SLL and B cell non-Hodgkin’s lymphoma	Recruiting
** Umbralisib (TGR-1202)	NCT04624633	Ublituximab/acabrutinib	Anti-CD20/BTK	2	Chronic lymphocyte leukemia	Not yet recruiting
** Umbralisib (TGR-1202)	NCT03828448	Ublituximab	Anti-cd20	2	Naive follicular lymphoma	Recruiting
** Umbralisib (TGR-1202)	NCT03776864	Pembrolizumab	Anti-PD1	2	Classical Hodgkin’s lymphoma	Recruiting
** Umbralisib (TGR-1202)	NCT03776864	Carfilzomib	Proteasome	1b	Refractory Hodgkin’s lymphoma	Recruiting
** Umbralisib (TGR-1202)	NCT02612311	Ublituximab	Anti-CD20	3	CLL	Active
** Umbralisib (TGR-1202)	NCT02656303	Ublituximab	Anti-CD20	2	CLL	Enrolling by invitation
** Umbralisib (TGR-1202)	NCT03207256	Ublituximab	Anti-CD20	Compassionate trial	B cell non-Hodgkin’s lymphoma or CLL	Enrolling by invitation
** Umbralisib (TGR-1202)	NCT04635683	Ublituximab/lenalidomide	Anti-CD20/cereblon	1	Non-Hodgkin’s lymphoma, mantle cell lymphoma	Not yet recruiting
*PI3K-delta/gamma* inhibitor						
** Duvelisib	NCT03892044	Nivolumab	Anti-PD1	1	Richter’s syndrome, follicular lymphoma	Recruiting
** Duvelisib	NCT04652960	Nivolumab	Anti-PD1	1	Sezary syndrome (T cell lymphoma)	Not yet recruiting
** Duvelisib	NCT04193293	Pembrolizumab	Anti-PD1	1b/2	Head and neck carcinoma	suspended
Tenalisib (RP6530)	NCT03471351	Pembrolizumab	Anti-PD1	1/2	Relapsed or refractory cHL	Terminated
*PI3K-delta* inhibitor						
** Idealisib	NCT03639324	Venetoclax	BCL2	1	Relapsed or refractory lymphocytic leukemia, small lymphocytic lymphoma	Recruiting
** Idealisib	NCT03639324	Rituximab	Anti-CD20	1	Relapsed or refractory lymphocytic leukemia, small lymphocytic lymphoma	Recruiting
** Idealisib	NCT02332980	Pembrolizumab	Anti-PD1	2	Relapsed or refractory lymphocytic leukemia, other low-grade B cell non-Hodgkin’s lymphomas	Recruiting
* Zandelisib	NCT04745832	Rituximab	Anti-CD20	3	Non-Hodgkin’s lymphoma, marginal lymphoma, follicular lymphoma	Not yet recruiting
INCB050465	NCT02646748	Itacitinib	JAK1	1b	Advanced solid tumor	Active, not recruiting
INCB050465	NCT02646748	Pembrolizumab	Anti-PD1	1b	Advanced solid tumor	Active, not recruiting
*PI3K-alpha/delta* inhibitor						
** Copanlisib	NCT0446271	Vemurafenib	Raf	1b	Thyroid cancer	Recruiting
** Copanlisib	NCT03502733	Nivolumab	Anti-PD1	1b	Advanced cancer and lymphoma	Recruiting
** Copanlisib	NCT03502733	Ipilimumab	Anti-PD1	1b	Advanced cancer and lymphoma	Recruiting
** Copanlisib	NCT04108858	Tratuzumab	Erbb2	1b/2	Advanced Her2-positive breast cancer	Recruiting
** Copanlisib	NCT04108858	Pertuzumab	ERbb2	1b/2	Advanced Her2-positive breast cancer	Recruiting
** Copanlisib	NCT03711058	Nivolumab	Anti-PD1	1/2	Relapsed, refractory, metastatic, or unresectable solid tumors or colorectal cancer	Recruiting
** Copanlisib	NCT04431635	Nivolumab/rituximab	Anti-PD1/anti-CD20	1b	Relapsed or refractory indolent lymphoma	Recruiting
** Copanlisib	NCT03789240	Rituximab	Anti-CD20	2	Untreated follicular lymphoma	Recruiting
** Copanlisib	NCT02367040	Rituximab	Anti-CD20	3	iNHL	Active, not recruiting
** Copanlisib	NCT04155840	Bendamustine/rituximab	DNA repair/anti-CD20	2	CLL, small lymphocytic lymphoma	Suspended
Pictilisib	NCT01493843	Bevacizumab/carboplatin/paclitaxel	Anti-EGFR	2	Advanced and recurrent non-small cell lung cancer	Completed
*PI3K-gamma* Inhibitor						
Eganelisib (IPI-549)	NCT03980041	Nivolumab	Anti-PD1	2	Advanced urothelial carcinoma	Active, not recruiting
*Pan-PI3K* inhibitor						
* Paxalisib (GDC0084)	NCT03765983	Trastuzunab	Anti-HER2	2	HER2-positive breast cancer	Recruiting
* Buparlisib (BKM120)	NCT02049541	Rituximab	Anti-CD20	1	Indolent B cell lymphoma	Active, not recruiting
* Buparlisib (BKM120)	NCT02049541	Paclitaxel	Received anti-PD1 therapy	3	Head and neck cancer	Recruiting
* Buparlisib (BKM120)	NCT01512251	Vemurafenib	Braf	1	Advanced melanoma	Completed
*PI3K-alpha* inhibitor						
** Alpelisib (BYL719)	NCT04208178	Trastuzumab	Anti-HER2	3	HER2-positive advanced breast cancer with PIK3CA mutation	Recruiting
** Alpelisib (BYL719)	NCT04208178	Pertuzumab	Anti-HER2	3	HER2-positive advanced breast cancer with PIK3CA mutation	Recruiting
** Alpelisib (BYL719)	NCT02167854	Trastuzumab	Anti-HER2	1	Metastatic HER2-positive breast cancer	Active, not recruiting
** Alpelisib (BYL719)	NCT02167854	Elgemtumab (LJM716)	Anti-HER2	1	Metastatic HER2-positive breast cancer	Active, not recruiting
** Alpelisib (BYL719)	NCT02390427	Trastuzumab	Anti-HER2	1	Advanced HER2+ breast cancer	Active, not recruiting
*PI3K-beta* inhibitor						
GSK2636771	NCT03131908	Pembrolizumab	Anti-PD1	1/2	Metastatic melanoma and PTEN loss	Active, not recruiting
*PI3K-alpha/delta/gamma* inhibitor						
Pilaralisib	NCT01042925	Trastuzumab/paclitaxel	Anti-HER2	1/2	Metastatic breast cancer	completed
*Akt* inhinbitor						
Cyclophosphamide	NCT03139370	KITE-718	MAGE-A3 and/or MAGE-A6	1	Solid tumor	Recruiting
Fludarabine	NCT03139370	KITE-718	MAGE-A3 and/or MAGE-A6	1	Solid tumor	Recruiting

* Fast track designation; ** FDA-approved.

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
