# Peer review of "PI3K/Akt Pathway: The Indestructible Role of a Vintage Target as a Support to the Most Recent Immunotherapeutic Approaches"

_cancers, 2021, doi:10.3390/cancers13164040_

Round 1

Reviewer 1 Report

The author summarized the role of PI3K inhibitors as immunomodulator and potential combinatory therapy with immunotherapeutic agents. The paper presents the connection between the PI3K/Akt pathway, which is commonly mutated across cancers, with immunomodulatory aspect of cancer and demonstrated current clinical studies being done with PI3K inhibitors. The topic of the paper is relevant to the aim and scope of the journal and nicely demonstrated current clinical studies going on with PI3K inhibitors. However, the scientific significance of PI3K/Akt pathway as a target for immunotherapies needs more in-depth content prior to introducing subsequent examples. In addition, the manuscript contains numerous errors and typos and must be reviewed.

  • In section 1.2 the title is PI3K/Akt inhibition and anti-tumor immunity, however, does not provide enough content for PI3K/Akt signaling and immunity for anti-tumor effect. The paragraph is mostly written on the CAR-T therapy and immune check point inhibitor without providing the connection to PI3K/Akt signaling. This section of introduction should provide more in-depth content relating to the title. In addition, section 1.2 is written with poor grammar, which is hindering for readers to understand.
  • The paper overall contains numerous spelling errors and minor typos throughout the manuscript. In addition, the words and nomenclature are not consistent throughout all sections. For example, section 3 uses zeta and section 4 uses delta and gamma word instead of the Greek symbols.
  • In section 3, line 330 to 335, the “T(naïve)N/T(stem 330 central memory)SCM and increased T(effector)EFF subsets” and line 356 (TN/TCM phenotype) seem to be in different nomenclature and must be clarified or consistent.
  • Many words contain a hyphen within a word and must be corrected. Example – line 28, 48 and many more.
  • Unnecessary capitals in line 138 (Colorectal Cancer), 153 (Melanoma), 173 (Non-Small Cell Lung Cancer), line 40 (Signaling) and many more.
  • Line 211, For examples is wrong grammar.
  • In line 308, AKT inhibitor is abbreviated as AKTi after repeated use of term “Akt inhibitor” in section 3. Suggest the abbreviation when the term first appears.

Reviewer 2 Report

This review describes that inhibition of PI3K/Akt pathway contributes to enhance anti-tumor immunity. Authors explain  well the subject including immunomodulatory drugs in combination with PI3K inhibitors, modulation of PI3K to improve CAR T therapy, PI3K/Akt inactivation during the CAR T-cells manufacturing and classification of immunomodulatory inhibitors used in clinical trials. This review could be useful for designing combi-therapy of anti-cancer drugs involved in PI3K/AKT or related signaling pathways. However, some mistakes, sug-gest, char-acterized should be corrected. 

Round 2

Reviewer 1 Report

The manuscript was properly revised based on the suggested comments.